# Perinatal Mental Illness in the Middle East and North Africa Region—A Systematic Overview

**DOI:** 10.3390/ijerph17155487

**Published:** 2020-07-29

**Authors:** Sathyanarayanan Doraiswamy, Anupama Jithesh, Sonia Chaabane, Amit Abraham, Karima Chaabna, Sohaila Cheema

**Affiliations:** Institute for Population Health, Weill Cornell Medicine—Qatar, Education City, Doha 24144, Qatar; anj2022@qatar-med.cornell.edu (A.J.); soc2016@qatar-med.cornell.edu (S.C.); ama2006@qatar-med.cornell.edu (A.A.); kac2047@qatar-med.cornell.edu (K.C.); soc2005@qatar-med.cornell.edu (S.C.)

**Keywords:** perinatal mental illness, antepartum depression, postpartum depression, Middle East and North Africa

## Abstract

*Aims:* Perinatal mental illness (PMI) is associated with a high risk of maternal and infant morbidity. Recently, several systematic reviews and primary studies have explored the prevalence and risk factors of PMI in the Middle East and North Africa (MENA) region. To our knowledge, there has been no critical analysis of the existing systematic reviews (SRs) on this topic in the MENA region. Our systematic overview primarily aimed to synthesize evidence from the published SRs on PMI in the MENA countries focusing on a) the prevalence of PMI and b) the risk factors associated with PMI. *Methods:* We conducted a systematic overview of the epidemiology of PMI in the Middle East and North Africa region by searching the PubMed, Embase, and PsycInfo databases for relevant publications between January 2008 and July 2019. In addition to searching the reference lists of the identified SRs for other relevant SRs and additional primary studies of relevance (those which primarily discussed the prevalence of PMI and/or risk and protective factors), between August and October 2019, we also searched Google Scholar for relevant studies. *Results:* After applying our inclusion and exclusion criteria, 15 systematic reviews (SRs) and 79 primary studies were included in our overview. Studies utilizing validated diagnostic tools report a PMI prevalence range from 5.6% in Morocco to 28% in Pakistan. On the other hand, studies utilizing screening tools to detect PMI report a prevalence range of 9.2% in Sudan to 85.6% in the United Arab Emirates. Wide variations were observed in studies reporting PMI risk factors. We regrouped the risk factors applying an evidence-based categorization scheme. Our study indicates that risk factors in the relational, psychological, and sociodemographic categories are the most studied in the region. Conversely, lifestyle-related risk factors were less studied. *Conclusions:* Our systematic overview identifies perinatal mental illness as an important public health issue in the region. Standardizing approaches for estimating, preventing, screening, and treating perinatal mental illness would be a step in the right direction for the region.

## 1. Introduction

Mental illness in women during the perinatal period (start of pregnancy until one year postpartum) can have a significant impact on maternal and infant morbidity [1,2] Perinatal mental illness (PMI) include depression, anxiety, and postpartum psychoses, the latter of which usually manifests as bipolar disorder [1]. Studies also report suicide as one of the major causes for maternal mortality in several countries [3,4]. The mental health of women during the perinatal period also has a profound impact on the well-being of the infant. Perinatal depression has been shown to be associated with higher rates of infant/child malnutrition and stunting; poor adherence to immunization schedules; increased susceptibility to infectious diseases, including diarrhea among infants and children [5]; epigenetic disruption (linked to nuclear receptor subfamily 3, group C, member 1 methylation), leading to poor physical and mental health outcomes in the offspring [6]; and poor cognitive, emotional, behavioral, and social development of children [2].

Systematic reviews (SRs) conducted in high-income countries estimate that about 10% of pregnant women and 13% of those in the postpartum period [7] are likely to experience some form of mental illness, the commonest being anxiety and depression [2]. Poorer perinatal mental health among women in low- and low–middle-income countries has been previously demonstrated [8]. The first ever systematic review reporting the prevalence of nonpsychotic mental illness among pregnant and postpartum women in 17 low- and low–middle-income countries by Fisher et al. estimated that 15.6% of antepartum women and 19.8% of postpartum women suffered from one or more nonpsychotic forms of mental illness [8].

The World Health Organization’s Commission on the Social Determinants of Health found that perinatal mental health is poorer among women of lower socioeconomic status and other marginalized women [9]. Negative gender-based factors, including all forms of gender-based violence [10], have been established to have a deleterious effect on perinatal mental health. Poor education, job insecurity, and lack of a trustworthy partner are also proven risk factors for PMI [8].

Recently, several systematic reviews and primary studies have explored the prevalence and risk factors of PMI in the Middle East and North Africa (MENA) region [11]. To our knowledge, there has been no critical analysis of the existing SRs on this topic in the MENA region. Our systematic overview primarily aimed to synthesize evidence from the published SRs on PMI in the MENA countries focusing on (a) prevalence of PMI and (b) risk factors associated with PMI. We restricted the overview to only PMI as the etiopathogenesis of mental illness in the perinatal period is quite different from that of the general female population [12]. We also aimed to summarize measurement variations in PMI, to identify research gaps, and to provide recommendations to improve the mental health of perinatal women in the region.

## 2. Materials and Methods

We conducted the overview in accordance with the Cochrane Handbook for Systematic Reviews for Interventions [13]. This overview is part of a series of research and publications aimed at synthesizing available literature on population health issues in the MENA region, assessing them for their quality with an aim of contributing to improving the quality of evidence generated in the region [14,15].

The overview draws from an a priori protocol registered with the International Prospective Register of Systematic Reviews (PROSPERO registration number CRD42017076736) [15,16]. We report this overview in line with the standards set in the Preferred Reporting Items for Systematic Reviews and Meta-Analyses (PRISMA) 2009 guidelines [17] and the Preferred Reporting Items for Overviews of Systematic Reviews (PRIO-harms) tool [18]. The PRISMA checklists for the review [19,20], overview [18], and the abstracts [21] have been enclosed as Appendix A, Appendix A, respectively.

### 2.1. Inclusion and Exclusion Criteria

We included SRs (inclusive publication dates: January 2008 to July 2019) of PMI from the MENA countries. We started our search from 2008 onwards as this was the year of publication of the first version of the Cochrane Handbook for Systematic Reviews for Interventions [13]. Manual search for other relevant articles was carried out between August and October 2019. The MENA countries included in the review were those in which Arabic, English, French, and/or Urdu are the official languages as the authors of this overview speak and read these languages. We considered all reviews as SRs, as long as they had a systematic approach to their search, including searching at least one database comprehensively with clear inclusion and exclusion criteria [15]. We restricted our search to only SRs which covered cross-sectional, case-control, and cohort designs. SRs which covered clinical trials and/or interventions were excluded.

### 2.2. Population of Interest

Our population of interest included pregnant and postpartum women up to one year after delivery living in any of the 20 MENA countries (namely, Algeria, Bahrain, Djibouti, Egypt, Iraq, Jordan, Kuwait, Lebanon, Libya, Morocco, Oman, Pakistan, Palestine, Qatar, Saudi Arabia, Sudan, Syria, Tunisia, the United Arab Emirates, and Yemen). These countries account for 8% of the global population [22]. According to the World Bank data of 2018, the crude birth rate in the MENA region was 23/1000 people [23].

### 2.3. Primary Outcomes

Our primary outcomes of interest included all epidemiological data from the SRs concerning perinatal anxiety; mania; bipolar disorder; postpartum blues; depression; psychoses, including schizophrenia; and psychiatric emergencies namely, self-inflicted injuries, suicides, and all other forms of mental illness. We also gathered conflicts of interest reported by the SR authors and that of the authors of the primary studies included in the SRs.

### 2.4. Literature Search and Data Management

Two reviewers (A.A., and S.C. (Sonia Chaabane)) searched three databases, namely PubMed, Embase, and PsycInfo. The search criteria were limited to reviews, systematic reviews, and meta-analyses concerning PMI. The search criteria used for each database is included as Appendix A, Appendix A. In addition to searching the reference lists of the identified SRs for other relevant SRs and additional primary studies of relevance (those which primarily discussed prevalence of PMI and/or risk and protective factors), between August and October 2019, we also searched Google Scholar for relevant studies during this period.

After removing duplicate publications with Endnote [24], independent title/abstract screening followed by independent full text screening were conducted by A.A., and S.C. (Sonia Chaabane), with Rayyan software [25,26]. Independent data extraction was carried out by A.A., and S.C. (Sonia Chaabane). Discrepant inclusions and extraction of SRs were discussed by A.A., S.C. (Sonia Chaabane), A.J., K.C., and S.D. under the supervision of the senior author.

### 2.5. Quality Assessment

The quality of the included SRs was assessed using the 11 criteria listed in the checklist of the measurement tool to assess the methodological quality of systematic reviews (AMSTAR) [27]. The quality of the primary studies reported in the SRs was assessed using the Population, Intervention, Comparison, Outcomes, Timing, Setting (PICOTS) framework [28] (interventions and comparators not being relevant in our overview).

### 2.6. Synthesis

All available data on the prevalence and risk factors associated with PMI of the population of interest was synthesized. We created evidence tables from the extracted data. Country level prevalence as estimated by different tools was separated for antepartum and postpartum women where possible. From the available studies, we also mapped the risk factors which have an impact on perinatal mental health. For the purpose of grouping the identified risk factors, we used a broad categorization scheme put forward by Furber et al. [29]. This categorization scheme lists all “ever” identified potential risk factors for mental illness under primary and secondary risk categories. We matched the risk factors identified by the primary studies in our overview with those in the categorization scheme and included them in the appropriate primary and secondary categories.

## 3. Results

Our initial search identified 11 SRs [8,30,31,32,33,34,35,36,37,38,39]. Our manual search identified an additional four SRs [40,41,42,43], making a total of 15 SRs on the epidemiology of PMI in at least one of the 20 MENA countries included in our overview. From the 15 SRs, we originally identified 134 primary studies. Twenty-eight primary studies [44,45,46,47,48,49,50,51,52,53,54,55,56,57,58,59,60,61,62,63,64,65,66,67,68,69,70,71] featured more than one SR. Six additional primary studies [52,72,73,74,75,76] relevant to the overview were identified by a manual search. All in all, 79 primary studies [44,45,46,47,48,49,50,51,52,53,54,55,56,57,58,59,60,61,62,63,64,65,66,67,68,69,70,71,72,73,74,75,76,77,78,79,80,81,82,83,84,85,86,87,88,89,90,91,92,93,94,95,96,97,98,99,100,101,102,103,104,105,106,107,108,109,110,111,112,113,114,115,116,117,118,119,120,121,122] from the 15 SRs, their reference lists, and those identified by additional manual search were included in the qualitative synthesis.

A PRISMA flowchart summarizing the search and inclusion of the systematic reviews and the primary studies is provided in Figure 1.

### 3.1. Characteristics of the Included SRs

Relevant data were available for 13 countries, namely Bahrain, Egypt, Jordan, Kuwait, Lebanon, Morocco, Oman, Pakistan, Qatar, Saudi Arabia, Sudan, Tunisia, and United Arab Emirates. Prevalence data were available for all countries except Kuwait. Twelve SRs [8,30,31,34,35,36,38,39,40,41,42,43] had prevalence data for one or more mental disorders such as anxiety, depression, postnatal blues, suicidal ideation, and suicides in perinatal women. Two SRs [33,37] focused exclusively on risk factors of PMI, and one SR focused on suicide [32] and its relative contribution to pregnancy-related mortality. One SR included risk factors from Kuwait [37]. All data extracted from the SRs and their primary studies are tabulated in detail in Appendix A, Appendix A.

### 3.2. Quality Assessment of the Included SRs

The quality of the SRs was assessed using the original version of the AMSTAR recommendations (Table 1) as it was found to be more appropriate for observational studies. While all SRs provided the list of included studies including their characteristics, none of them provided the list of excluded studies. No SR reported conflicts of interest in their included studies. Thirteen SRs (86.7%) [8,30,31,32,33,34,35,36,37,38,40,42,43] had a comprehensive literature search (defined as searching at least two databases). Five of them (33.3%) [30,32,33,36,38] actively looked for grey literature. Only three SRs (20%) [32,36,38] deployed two persons in data extraction (either independent extraction or one screen and another check procedure) and had a consensus procedure in place for disagreements. Four SRs (26.7%) [8,32,36,38] had included a publication bias assessment.

We recorded the conflicts of interest reported by the authors of the SRs. The authors of nine out of 15 SRs reported no conflict of interest [8,32,33,34,35,36,37,38,40]. Four declared their funding sources [8,32,33,37], and two explicitly declared “no external funding” [36,40]. Six of the SRs did not declare the conflict of interest of its authors [30,31,39,41,42,43]. The sources of funding for nine SRs [30,31,34,35,38,39,41,42,43] are unknown.

### 3.3. Overview of SRs with Prevalence Data

From the 15 included SRs, prevalence of the common perinatal mental disorders in 12 countries (previously listed) has been reported in Table 2. Among the SRs, suicide contributing to overall pregnancy-related mortality was discussed in only one SR [32], which included data from four countries namely Jordan, Tunisia, Pakistan, and Egypt. Among the 11 SRs [8,30,31,34,35,38,39,40,41,42,43] providing prevalence data on postpartum mental illness, seven SRs [30,31,34,35,38,42,43] focused on the postpartum period only and four [8,39,40,41] covered both the antepartum and postpartum periods.

### 3.4. Suicidal Ideation and Suicides

The SR on suicide contribution to pregnancy-related mortality [32], through its meta-analysis, estimates 0.4% (95% CI: 0.1–0.9) contribution to pregnancy-related deaths for the two countries in the region (Tunisia and Egypt) where specific data is available. This ranged from 0.3% in Egypt [107] (95% CI: 0.04 to 1.1) to 0.6% (95% CI: 0.1–1.9) in Tunisia [90]. When the numerator is changed to include suicides, falls, drowning, poisoning, and burns, the average proportional contribution to pregnancy-related deaths becomes 3.5% (95% CI: 0.4–9.4) for Jordan, Egypt, and Tunisia. The country specific estimates range from a low of 0.6% (95% CI: 0.1–1.9) in Tunisia [90] to a high of 6.2% in Jordan [122] (95% CI: 2.5 to 12.4) with 5.8% (95% CI: 4.1–7.9) estimated in Egypt [107]. The authors acknowledge that these proportions are likely to be underestimates because of underreporting and nonrecognition of suicides as causes of pregnancy-related deaths in eligible studies.

There was only one other SR [39] comprising one primary study [82] from Pakistan which discussed suicide, suicidal ideation, and suicide attempts. The study reported that 11% of pregnant women screened between 20–26 weeks of gestation had suicidal ideation; 45% of those or 5% of pregnant women overall had attempted suicide in that study. None of the other SRs discussed suicidal ideation and suicides as part of their review of PMI.

### 3.5. Overview of SRs with Data on Risk Factors

Twelve of the 15 SRs [8,30,31,33,34,35,37,38,39,40,42,43] discussed risk factors for PMI. Risk factors were discussed as “risk factors”, “influencing factors”, and “associated factors” in the SRs. None of the SRs attempted to find factors associated exclusively with the antepartum period. Seven SRs [30,31,34,35,38,42,43] mapped risk factors for the postpartum period, and five SRs [8,33,37,39,40] discussed risk factors associated with the entire perinatal period. Except for one SR [39] which focused exclusively on Pakistan, all the other SRs attempted to cover a broader set of countries grouped as “Arab world”, “Arab Middle East”, and “Asian cultures”.

### 3.6. Quality Assessment of the Primary Studies

The primary studies included in the 15 SRs for this overview were assessed using the PICOTS framework. A summary of the quality of the primary studies is presented in Table 3, and a detailed listing is provided as Appendix A, Appendix A.

We identified that all SRs consistently reported the population covered by the respective primary studies. However, the exact point in the perinatal period was clearly defined in 44 (55.5%) studies only [10,44,45,46,47,48,49,50,51,52,53,54,55,56,57,58,59,60,61,62,63,65,66,67,68,70,71,77,81,83,86,89,90,95,97,98,101,107,111,112,115,117,118,121]. Mentioning the exact point of measurement is important, for example, to distinguish serious mental illness from postpartum blues, which are often mild and appear only in the first few weeks after delivery. Among the 73 primary studies included in the SRs, the timing, defined as at least the period (year and months) in which the study was carried out, was explicitly mentioned in four studies (5.5%) only [90,97,107,121]. We believe that this is a serious omission because of the possibility of seasonal affective disorders in perinatal women [122].

The setting defined as clinic-based or population-based was indicated in 50 studies (69.4%) only [44,45,48,49,53,54,56,57,58,59,61,62,63,64,66,67,69,70,77,78,79,80,81,82,83,84,85,86,88,89,92,95,97,98,99,101,104,105,106,107,108,109,111,112,115,116,117,118,119,123].

Non-reporting of the setting introduces bias within the sample as it fails to distinguish between clients with advanced conditions who report to a health facility when compared to the milder forms and/or those with poor health-seeking behaviors who may not report to a health facility.

All six additional primary studies (not mentioned in the SRs but identified by manual search) [72,73,74,75,76,120] clearly defined the population of interest. However, only three of them [74,75,76] defined the specific time period of the study and three [73,75,76] clarified the study setting.

Though not a usual part of the quality assessment, it is worthwhile to reflect on the validity of the tools used for estimating prevalence in the primary studies. With the exception of the Aga Khan University Anxiety and Depression Scale (AKUADS), all other tools (screening and diagnostic) were developed using samples from western settings. Tsai et al. have discussed this issue in the context of African settings [124]. They discuss an “etic approach” in which the construct of mental illness is promoted irrespective of cultures and an “emic approach” which emphasizes on the evaluation of mental illness constructs within a specific cultural context. The fields of mental illness and perinatal illness have long advocated for the need for integrating the etic and emic validation criteria to obtain more reliable prevalence estimates and to study risk factor associations [125]. We did not find a discussion on the use of such a hybrid approach in data collection in any of the primary studies included in the SRs. Four of the primary studies in our overview used the Arabic and Urdu version of the EPDS, but their validation process is unknown [54,75,101,103].

### 3.7. Overview of Primary Studies with Prevalence Data

Twenty-one [10.51,55,58,60,65,67,68,70,75,76,82,86,89,95,98,111,112,115,117,120] and thirty-two [45,47,48,49,51,52,53,54,56,57,58,59,63,65,66,67,69,72,74,78,81,83,94,101,103,104,108,109,110,113,119,120] primary studies reported prevalence data for the antepartum period and postpartum period, respectively.

#### 3.7.1. Antepartum

Antepartum mental illness prevalence data reported in 21 primary studies came from three countries only (Pakistan, N = 19 [10,55,58,60,67,68,70,75,76,82,86,89,95,98,111,112,115,117,118]; Jordan N=1 [51], and Morocco N = 1 [65]). Nineteen out of 21 (90.5%) studies [10,51,55,58,60,65,67,68,70,75,76,86,89,95,111,112,115,117,118] focused on depression. One study [83] provided a combined prevalence of anxiety and depression, and one study [98] chose to label the illness studied as “perinatal mental disorders”.

Sixteen out of 21 (76.2%) studies [10,51,55,58,65,70,75,76,82,86,95,111,112,115,117,118] were conducted in hospitals, four were exclusively from community settings [60,67,68,98], and one included both hospital and community-based samples [70]. The majority of the studies (52.3%, 11/21) [10,51,68,70,76,89,111,112,115,117,118] included pregnant women from all three trimesters. Most studies in general did not distinguish between screening and diagnostic tools in estimating prevalence.

The tool most commonly used in the primary studies (N = 7) [55,58,65,68,75,95,112] to estimate the prevalence of mental illness during the antepartum period was the Edinburgh Postnatal Depression Scale (EPDS). The EPDS cutoffs varied widely and ranged from 10–13 as opposed to the standard cutoff of 13 [126]. Four studies [60,83,99,115] used the Aga Khan University Anxiety Depression Scale (AKUADS), four studies [82,90,117,118] used the Hospital Anxiety and Depression Scale (HADS), and two studies [70,76] used the Centre for Epidemiology studies—Depression (CES-D) scale. The Mini International Neuropsychiatric Interview (MINI) [51]; World Health Organization (WHO) schedule for clinical assessment in neuropsychiatry (SCAN) [67]; Depression, Anxiety, and Stress Scale (DASS-42) [86]; and Hamilton depression scale (HAM-D) [111] were used by one study each. The SCAN, MINI, and HAM-D are the only recognized diagnostic tools (administered by trained psychologists/psychiatrists), whereas all other utilized tools used are screening tools.

With the various screening tools used in the region, we found the antepartum mental illness prevalence to range from 11.5% in a community setting (measuring all mental disorders using AKUADS) [98] to 75% (measuring depression using EPDS) [95] in a hospital setting in Pakistan. The only study from Jordan [65] utilizing the EPDS in a hospital setting estimated a prevalence of 19% antepartum depression. We noted wide variations in the cutoffs used in various studies and hence the “prevalence”, leading to difficulties in interpreting the reported data. Restricting the data reported from the use of diagnostic tools in Pakistan, a community-based study by Rahman et al. [67] using SCAN estimated the prevalence of all forms of antepartum mental illness to be 25%, and the study by Sadaf et al. [111] estimated a prevalence of 10% antepartum depression in a hospital-based sample using HAM-D. In Morocco, Alami et al. [51] estimated a prevalence of 19.2% antepartum depression also in a hospital setting, using MINI.

#### 3.7.2. Postpartum

Thirty-two primary studies [45,47,48,49,51,52,53,54,56,57,58,59,63,65,66,67,69,72,74,78,81,83,94,101,103,104,108,109,110,113,119,120] from 12 countries provided prevalence data during postpartum period. In addition to postpartum depression, two studies [10,124] included anxiety in their measurement, two studies [95,111] included postpartum blues, and two others combined all as “perinatal disorders” [67,98]. Twenty-three of the 32 primary studies (71.8%) used the EPDS [45,47,48,49,51,52,53,54,56,57,58,59,63,65,69,74,78,101,103,104,113,119,120] to measure postpartum depression. Six studies [47,51,56,72,83,101] used MINI. Four studies [8,66,67,108] used SCAN, and two other studies [45,109] used the World Health Organization self-reporting questionnaire (WHO SRQ-20). One study [123] used AKUADS, and the other [94] used Morsbach/Gordon maternity blues questionnaire and Pitt’s questionnaire. The tool used in one study was unclear [110].

Of the 32 studies, 9 measured postpartum depression at less than 6 weeks postpartum [45,47,49,51,59,63,66,94,120], 12 of them provided data during the six week to 3 month period [48,49,51,52,54,56,57,63,65,66,67,101], and 20 studies [47,51,53,54,58,65,66,69,72,78,83,94,103,104,108,109,110,113,119,123] in the three month to one-year period and one study [74] provided prevalence for the <6 week to 3 month period. Seven of these studies [49,51,53,54,65,66,101] had a prevalence at more than one point, and two among them [51,66] included all three points (<6 weeks, −3 months, and −1 year). There was a wide range of prevalence among the studies based on the instrument used, time of measurement in the postpartum period, and the setting.

Prevalence data on diagnostic interviews using MINI and SCAN were available for four countries only, namely Morocco, Pakistan, Saudi Arabia, and Sudan. However, the prevalence of postpartum depression in Sudan using MINI [101] was obtained from a two-stage sampling design with pre-diagnostic screening by EPDS and hence cannot be seen as true prevalence. Similarly, in Pakistan, the study of Rahman and Creed [66] has reported a postpartum depression prevalence range of 62% to 95% during different time points in the postpartum period. However, these proportions are for those women who were already diagnosed with depression during the antenatal period. This prevalence reported in the SRs hence cannot be taken as true prevalence. The postpartum depression prevalence measured using MINI in a hospital settings varied between 5.6% in Morocco [47] and 10.2% in Saudi Arabia [84] during the time period from 3 months to 1 year postpartum. In Pakistan, the prevalence of combined postpartum mental disorders varied between 25% from 6 weeks to 3 months [108] to 28% [67] from 3 months to 1 year postpartum, using the WHO SCAN. With the EPDS, the range of postpartum depression varied widely between 9.2% in Sudan [101] to 85.6% in United Arab Emirates (UAE) [56].

### 3.8. Overview of Primary Studies with Data on Risk Factors

A total of 53 primary studies [44,45,46,47,48,49,50,51,52,53,54,55,56,57,58,59,60,61,62,63,64,65,66,67,68,69,70,71,72,74,76,77,79,80,82,83,84,85,87,88,89,91,92,93,103,104,105,106,113,114,119,123,127] from 12 MENA countries (Bahrain, Egypt, Jordan, Kuwait, Lebanon, Morocco, Oman, Pakistan, Qatar, Saudi Arabia, Tunisia, and United Arab Emirates) reported data on risk factors. Data from low-income countries in the MENA region, notably Sudan, Yemen, and Djibouti, are missing. Data were also missing from some of the worst conflict-affected countries of the region such as Palestine, Syria, and Iraq.

We grouped the available risk factor data and provided a summary in Table 4. We have attempted to summarize the available odds ratio and relative risk reported by the various studies in the summary table. The majority of the studies which reported odds ratio and relative risk were cross-sectional studies. They had not been designed to assess risk factors. SRs reporting these studies have not provided an adequate description of the statistical analysis in the primary studies to determine whether the identified risk factors are independently associated with the outcome. They were then considered as potential risk factors as their independence has not been ascertained. Given the measurement issues (choice of instrument, cutoff points, and lack of theoretical framework), any attempt to study association without controlling variables by individual studies is a major barrier in deriving quality evidence [128,129]. Studies in the region continue to study relational (spousal and others), psychological, and sociodemographic factors in detail while overlooking lifestyle, environmental and occupational factors. Detailed information on the various factors as extracted from the primary studies are detailed in Appendix A, Appendix A.

Only a limited number of primary studies (N = 20) [44,46,49,50,51,55,57,58,59,60,61,65,67,68,70,72,76,77,82,105] had related data during the antepartum period to find associated factors. All SRs reported that familial social support (particularly spouse and/or mother-in-law) was positively associated with antepartum mental health. There was no consensus surrounding the association between education, employment, and financial situation and antepartum mental health. Postpartum mental health data was available from 43 studies [45,47,48,49,51,52,53,54,56,57,58,59,62,63,64,65,66,67,69,71,72,74,79,80,83,84,85,86,88,89,91,92,93,103,104,105,106,113,114,119,123,127], and the SRs were uniform in concluding the positive association between spousal support and postpartum mental health. Education levels and financial stress were found to be negatively associated with postpartum mental health.

While individual studies had identified associations between postpartum mental health and antepartum depression, stressful life events, mode of delivery, wanted/unwanted pregnancy, number of children, age at marriage, breast-feeding practice, and health of the infant, all SRs found very few studies of good quality to generate conclusive evidence.

This section may be divided by subheadings. It should provide a concise and precise description of the experimental results, their interpretation, as well as the experimental conclusions that can be drawn.

## 4. Discussion

Our systematic overview synthesizes available data on PMI prevalence and the associated risk factors. The overview points out that PMI is a major public health issue in the region. In the MENA region, there is limited data which are diagnostic in nature to be able to reliably assess country level prevalence. The published SRs and associated primary studies have measured various types of mental illness at different times during pregnancy and the postpartum period; have utilized different screening scales; and, even when using the same scales, have used different cutoff points, all of which makes the data highly heterogenous and difficult to compare. The SRs with meta-analyses in our overview have not questioned the use of screening tools for determining prevalence and have not taken into consideration the sensitivity and specificity of the respective screening tools in attempts to better estimate true prevalence. Our concern is shared by other researchers [128,129] who have questioned the use of self-reporting screening tools for prevalence estimation. A standardized framework to group or to clearly define risk factors in the studies is also lacking. The region’s SRs on the topic have been of varying quality, with all SRs consistently omitting to report on the conflicts of interest in individual studies or to present a list of excluded studies and with most of them failing to discuss publication bias as recommended by international guidelines. Few published SRs follow the recommended procedures for data extraction and resolving disagreements in inclusion of studies. The majority of the SRs do not analyze/discuss the individual studies using the PICOTS framework, making it difficult to understand if PICOTS items have not been reported by the SRs or if the individual studies themselves did not carry out their research accordingly or report their approach explicitly.

However, the data available from the SRs and the included individual studies provide key insights on perinatal mental health in the region. The SR of Fuhr et al. [32] helps provide context to this discussion. The MENA region that our overview discusses includes most countries in the Eastern Mediterranean (EMRO) region as defined by the World Health Organization and as adopted by Fuhr et al. [32]. It has been found that the EMRO region has a lower than average proportion of global pregnancy-related deaths attributable to suicide but a higher than global average proportion of pregnancy-related deaths due to injuries in general or when deaths due to suicide, falls, drowning, poisoning, and burns are combined together. This discrepancy is unique to the EMRO region, implying an underreporting of suicides. Preexisting mental illness is a risk factor for suicides; underreporting of suicides may be due to prevailing stigma towards suicides and its associated underlying mental illness. The fact that other SRs did not attempt to systematically include and discuss suicides in their scope of study indicates either a low recognition of suicides as a manifestation of perinatal illness and/or a lack of data to study this in depth.

The SRs which provide data on the prevalence of mental illness focus heavily on depression in the perinatal period. Antepartum data based on diagnostic criteria are available from Morocco and Pakistan only, making it difficult to generalize the findings for the region. For the postpartum period though, meaningful diagnostic data was available for only three countries (Morocco, Pakistan, and Saudi Arabia) with a wide-ranging prevalence. We also note that the diagnostic tools used different definitions for depression/depressive disorders in particular due to the changes in International classification of diseases (ICD) and Diagnostic and Statistical Manual of Mental Disorders (DSM) classification over the last decade. Due to these shortcomings, making any interregional/national comparison presents a challenge. However, in general, the finding of high prevalence in some countries of the region is of concern and of public health importance. This reported prevalence in these countries meanwhile needs to be seen in the context of the risk factors identified in the overview.

Very few primary studies included in the SRs outline associated factors with antepartum mental illness. However, for postpartum mental illness, specifically depression, the SRs included in the review consistently found positive association with social support (from spouse and in-laws), education levels, and financial stability. Recently, more women in the MENA region are completing higher education and are entering the labor market but are still expected to maintain their central role in managing the family [11]. Under these circumstances, family support becomes extremely critical for women to be able to maintain optimal mental health. Education and wealth are likely to be factors which prevent the women from falling into a vicious cycle of stress during pregnancy and after giving birth. There is a need for more research in the region exploring risk factors for perinatal mental health related to lifestyle, occupation, and environmental exposures.

Our overview has important implications for practice and future research. Firstly, our overview highlights the lack of a standardized methodology to estimate the prevalence of PMI in the region. This may prevent appreciation of the true magnitude of the problem and may make comparison between and within countries difficult. Such comparisons could help countries understand the effectiveness of their identified approaches to addressing PMI. The research capacity on PMI should be strengthened by regional institutions such as the World Health Organization in partnership with leading academic institutions in the region. Secondly, our overview provides synthesis of available data on PMI for individual countries in the region. This data can enable countries to recognize PMI as a significant public health problem and can help them develop evidence-based locally informed national guidelines for preventing, screening, and treating PMI. Countries with no available data should identify this as a major gap which must be addressed at the national level. Researchers from these countries should be encouraged to conduct good quality research on PMI. Also, regarding the potential risk factors associated with PMI, certain categories appear to be over-studied. Future research in the region should focus on risk factor categories which have either been poorly studied or not studied at all.

### Limitations

This overview has limitations. The data from primary studies were extracted from the SRs, and quality was assessed based on their findings as reported in the SRs. Extreme variations in PMI definitions, methods of assessment, and study quality precluded us from performing a meta-analysis to arrive at a pooled prevalence. The primary studies were reviewed only if outliers or errors were suspected. The studies included in our analysis have come from only 13 out of the 20 MENA countries, and the majority of these have been reported from one country: Pakistan. We were not able to gather any data for the relatively low-income countries of the region. Though we scanned extensively for both published and unpublished literature, all retrieved studies were in English, which is also acknowledged as a limitation.

## 5. Conclusions

There are very few SRs or primary studies with sufficiently good quality data of diagnostic nature from the MENA region to arrive at an informed prevalence of PMI. There is scope for strengthening the quality of future primary studies and SRs. Our data suggests that PMI is a major public health issue in the region. Social influence, particularly spousal support, is seen as a key aspect in improving perinatal mental health in the region. Programs and policies in raising awareness of husbands and in-laws through mass media and community gatherings as well as during antepartum and postpartum consultations can be useful interventions. Due to the low priority given to mental health, particularly suicides during pregnancy and the postpartum period, there appears to be low awareness and a high level of stigma associated with this. Additional efforts are needed to better capture data on perinatal suicides and other self-inflicted injuries in the region to be able to develop evidence-guided population-based policies.

## Figures and Tables

**Figure 1 ijerph-17-05487-f001:**
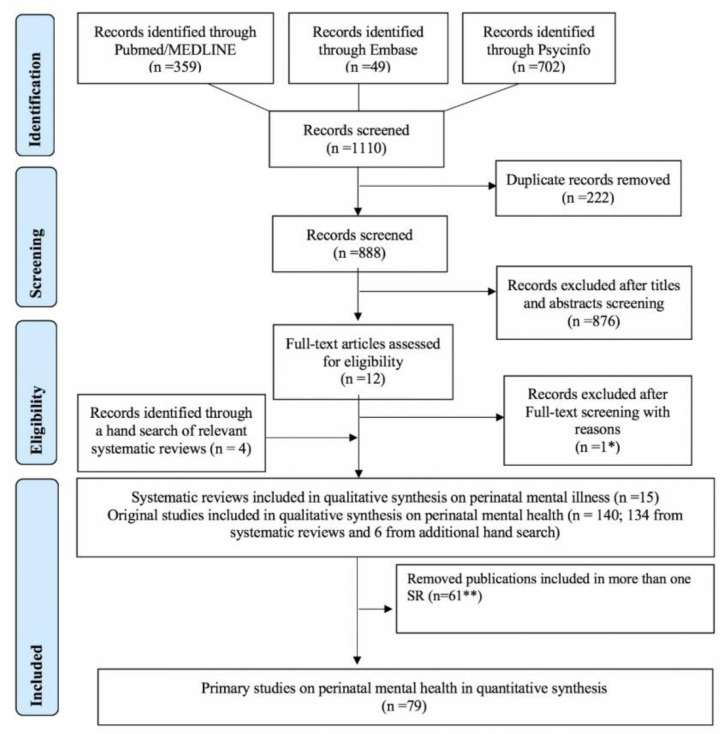
Preferred Reporting Items for Systematic Reviews and Meta-Analyses (PRISMA) 2009 flowchart of the systematic review’s inclusion. SR—Systematic reviews.

**Table 1 ijerph-17-05487-t001:** Quality assessment of the systematic reviews using AMSTAR 1.

Systematic Review	A Priori Design Included?	Duplicate Study Selection/Data Extraction?	Comprehensive Literature Search Performed?	Status of Publication as Inclusion Criteria?	List of Included Studies Provided?	Characteristics of Included Studies Provided? *	Quality of Studies Assessed and Documented?		Quality Assessment Used Appropriately?	Methods Used to Combine Results Appropriate?	Likelihood of Publication Bias Assessed?	Conflict of Interest Stated? **
(Afzal and Khalid, 2016) [30]	−	−	+	+	+	+		+	−	-	−	−
(Alhasanat and Fry-McComish, 2015) [42]	−	+	+	−	+	+		−	−	-	−	−
(Amber Haque, 2015) [31]	−	N/A	+	−	+	+		−	−	-	−	−
(Evagorou et al., 2016) [43]	−	−	+	−	−	−		−	−	-	−	−
(Fisher et al., 2012) [8]	−	−	+	−	+	+		+	+	+	+	+
(Fuhr et al., 2014) [32]	+	+	+	+	+	+		+	+	+	+	+
(James-Hawkins et al., 2019) [33]	+	+	+	+	+	+		+	+	-	N/A	+
(Jones and Coast, 2013) [34]	−	−	+	-	+	+		−	−	-	−	+
(Klainin and Arthur, 2009) [35]	+	−	+	−	+	+		−	−	-	−	+
(Mahendran et al., 2019) [36]	N/A	+	+	+	+	+		+	+	+	+	+
(Qutteina et al., 2018) [37]	−	+	+	-	+	+		+	+	-	N/A	+
(Sawyer et al., 2010) [40]	−	N/A	+	−	+	+		−	−	-	−	+
(Shorey et al., 2018) [38]	+	+	+	+	+	+		+	+	+	+	+
(Stewart, 2007) [41]	−	−	−	−	+	+		−	−	-	−	−
(Zahidie and Jamali, 2013) [39]	−	−	−	−	+	+		−	−	-	−	−

Categories for risk of bias are as follows: +, yes; −, no; N/A, not applicable. * No single review provided a list of excluded studies; ** conflict of interest of all included studies was never stated.

**Table 2 ijerph-17-05487-t002:** Reported prevalence of perinatal mental illness in the Middle East and North Africa.

Country	Study	Illness	Study Setting	Sample Size	Study Instrument	Instrument Cut off Score	Antepartum				Postpartum		
							1st Trimester	2nd Trimester	3rd Trimester	All Three	<6 Weeks	6 Weeks to 3 Months	>3 Months up to 1 Year
Bahrain	(Al Dallal and Grant, 2012) [48]	Depression	Hospital	237	EPDS	≥12	-	-	-	-	-	37.1%	-
Egypt	(Naglaa A. Mohamed and Maklof, 2011) [104]	Depression	Hospital	110	EPDS	Unclear	-	-	-	-	-	-	51.8%
Egypt	(Saleh el et al., 2013) [113]	Depression	Unclear	120	EPDS	≥13	-	-	-	-	-	-	17.9%
Jordan	(Mohammad et al., 2011) [65]	Depression	Hospital	353	EPDS	Unclear	-	-	19.0%	-	-	22.1%	21.2%
Jordan	(Yehia et al., 2013) [69]	Depression	Hospital	300	EPDS	Unclear	-	-	-	-	-	-	83.0%
Lebanon	(Chaaya et al., 2002) [53]	Depression	Hospital	396	EPDS	>12	-	-	-	-	-	-	21.0%
Morocco	(Agoub et al., 2005) [47]	Depression	Hospital	144	MINI	N/A	-	-	-	-	6.9–18.7%	-	5.6–11.8%
				144	EPDS	>12	-	-	-	-	20.1%	-	-
Morocco	(Alami et al., 2006) [51]	Depression	Hospital	100	EPDS	Unclear	-	-	-	-	21%	-	-
				100	MINI	N/A	17.4%	16.0%	15.7%%	19.2%	16.8%	14.0%	6–12%
Oman	(Al Hinai and Al Hinai, 2014) [49]	Depression	Hospital	282	EPDS	>13	-	-	-	-	13.5%	10.6%	-
Pakistan	(Ahmed, 2005) [78]	Depression	Hospital	90	EPDS	Unclear	-	-	-	-	-	-	27.0%
Pakistan	(Ali et al., 2009) [80]	Depression& Anxiety	Community	Unclear	AKUADS/ Unclear diagnostic tool	Unclear	-	-	-	-	-	-	28.8% *
Pakistan	(Ali et al., 2012) [81]	Depression	Hospital	167	HADS	≥8	-	-	-	16.8%	-	-	-
Pakistan	(Din et al., 2016) [86]	Depression	Hospital	230	DASS-42	≥9	-	-	29.1%	-	-	-	-
Pakistan	(Gulamani et al., 2013) [74]	Depression	Hospital	214	EPDS	Unclear	-	-	-	-	15.3–35.3%	-
Pakistan	(Habib, 1997) [94]	Postnatal blues & PPD	Unclear	30	Gordon’s (1984); Pitt’s (1968)	Unclear	-	-	-	-	50.0%	-	20–37%
Pakistan	(Fareeha Hamid, 2008) [89]	Depression	Hospital	100	HADS	Unclear	-	-	-	18.0%	-	-	-
Pakistan	(Humayun et al., 2013) [95]	Depression	Hospital	506	EPDS	≥ 10	-	-	75.0%	-	-	-	-
Pakistan	(Husain et al., 2006) [57]	Depression	Community	149	EPDS	≥12	-	-	-	-	-	36.0%	-
Pakistan	(Husain et al., 2011) [58]	Depression	Hospital	149	EPDS	>12	-	-	25.8%	-	-	-	38.3%
Pakistan	(Husain et al., 2014) [75]	Depression	Hospital	1357	EPDS	≥12	-	-	13.4%	-	-	-	-
Pakistan	(Haider, 2010)[55]	Depression	Hospital	213	EPDS	>12	-	-	42.7%	-	-	-	-
Pakistan	(Kalyani et al., 2001) [59]	Depression	Community	120	EPDS	>10	-	-	-	-	63.3%	-	-
Pakistan	(Karmaliani et al., 2006) [98]	Perinatal mental disorders	Community	1000	AKUADS	Unclear	-	11.5%	-	-	-	-	-
Pakistan	(Karmaliani et al., 2009) [60]	Depression	Community	1368	AKUADS	≥13	-	18.0%	-	-	-	-	-
Pakistan	(Kazi et al., 2006) [76]	Depression	Hospital	292	CES-D	≥16	-	-	-	39.4%	-	-	-
Pakistan	(Sharifa Mir, 2012) [115]	Depression	Hospital	328	AKUADS	>13	-	-	-	33.8%	-	-	-
Pakistan	(Muneer et al., 2009) [103]	Depression	Unclear	154	EPDS	Unclear	-	-	-	-	-	-	33.0%
Pakistan	(Asad et al., 2010) [83]	Depression/ Anxiety	Hospital	1368	AKUADS	Unclear	-	18.0%	-	-	-	-	-
Pakistan	(Syeda Rabia, 2017) [117]	Depression	Hospital	520	HADS	>8	-	-	-	23.1%	-	-	-
Pakistan	(Rahman et al., 2003)[67]	Perinatal mental disorders	Community	T0 = 701 T1 = 632 T2 = 541	SCAN	N/A	-	-	25.0% (T1)	-	-	28.0% (T2)	-
Pakistan	(Rahman et al., 2004a) [108]	Depression	Community	265	SCAN	N/A	-	-	-	-	-	-	25.0%
Pakistan	(Rahman et al., 2004b) [109]	Depression	Hospital	172	SRQ-20	Unclear	-	-	-	-	-	-	41.0%
Pakistan	(Rahman and Creed, 2007) [66]	Depression	Community	T1: 701 (antepartum) T2: 632 T3: 160 T4: 129	SCAN	N/A	-	-	-	-	94.0% (T2)	76.0% (T3)	62% (T4)
Pakistan	(Rasheed, 1988) [110]	Postnatal blues & PPD	Unclear	103	Unclear	Unclear	-	-	-	-	-	-	54.4%
Pakistan	(Sadaf, 2011) [111]	Depression	Hospital	150	HAM-D	Unclear	-	-	-	10.0%	-	-	-
Pakistan	(Saeed et al., 2016) [112]	Depression	Hospital	82	EPDS	≥9	-	-	-	42.7%	-	-	-
Pakistan	(Shah et al., 2011) [68]	Depression	Community	128	EPDS	≥13	-	-	-	46.9%	-	-	-
Pakistan	(Yasmeen et al., 2010) [119]	Depression	Hospital	100	EPDS	Unclear	-	-	-		-	-	41.0%
Pakistan	(Waqas et al., 2015) [118]	Depression	Hospital	500	HADS	≥11	-	-	-	31.8%	-	-	-
Pakistan	(Zahidie et al., 2011) [70]	Depression	Mixed	375	CES-D	≥16	-	-	-	61.1%	-	-	-
Qatar	(Bener et al., 2012a)[52]	Depression	Hospital	1379	EPDS	>12	-	-	-	-	-	17.6%	-
Saudi Arabia	(Al-Modayfer et al., 2015) [121]	Depression	Hospital	571	EPDS	Unclear	-	-	-	-	14.0%	-	-
Saudi Arabia	(Amr and Hussein Balaha, 2010) [72]	Depression	Hospital	367	MINI	N/A	-	-	-	-	-	-	6.0%
Saudi Arabia	(Balaha et al., 2009) [83]	Depression	Hospital	800	MINI	N/A	-	-	-	-	-	-	10.2%
Sudan	(Khalifa et al., 2015) [101]	Depression	Hospital	238	EPDS	≥12	-	-	-	-	-	9.2%	-
				40	MINI	N/A	-	-	-	-	-	45%*	-
Tunisia	(Masmoudi et al., 2008) [63]	Depression	Hospital	213	EPDS	>10	-	-	-	-	19.2%	13.2%	-
UAE	(Abou-Saleh and Ghubash, 1997) [45]	Depression	Hospital	95	EPDS	>11	-	-	-	-	18.0%	-	-
				95	SRQ	Unclear	-	-	-	-	24.0%	-	-
UAE	(Green et al., 2006) [54]	Depression	Hospital	T0 = 125 T1 = 86 T2 = 56	EPDS	≥13	-	-	-	-	-	22.0% (T1)	12.5% (T2)
UAE	(Hamdan and Tamim, 2011) [56]	Depression	Hospital	137	MINI	N/A	-	-	-	-	-	10.0%	-
					EPDS	>10	-	-	-	-	-	85.6%	-

N/A, not applicable as this is a diagnostic tool; * two stage sample and not true prevalence; T, time frame; EPDS, Edinburgh Postnatal Depression Scale; CES-D, Centre for Epidemiologic Studies Depression Scale; MINI, Mini International Neuropsychiatric Interview; SCAN, WHO Schedules for Clinical Assessment in Neuropsychiatry; AKUADS, Aga Khan University Anxiety and Depression Scale; HAM-D, Hamilton Depression Rating Scale; HADS, Hospital Anxiety and Depression scale; SRQ-20, Self-Reporting Questionnaire; and DASS, Depression, Anxiety, Stress Scale. In studies where prevalence was measured at multiple points of the perinatal period: T1= antepartum prevalence; T2 = 1st recording postpartum; T3 = 2nd recording post-partum and T4 = 3rd recording postpartum.

**Table 3 ijerph-17-05487-t003:** Summary of the quality assessment of primary studies included in the systematic reviews.

Attribute Defined	Population N (%)	Outcome N (%)	Data Collection Time N (%)	Study Setting N (%)
Yes	73 (100%)	73 (100%)	4 (5.5%)	50 (68.4%)
No	-	-	34 (46.6%)	15 (20.5%)
Unclear	-	-	35 (47.9%)	8 (9.6%)

**Table 4 ijerph-17-05487-t004:** Number of studies exploring various potential risk factors for perinatal mental illness.

Risk Categories *	Number of Studies													Odds Ratio/Relative Risk **		
	Total	Bahrain	Egypt	Jordan	Kuwait	Lebanon	Morocco	Oman	Pakistan	Qatar	Saudi Arabia	Tunisia	UAE	Low (95% CI)	Median	High (95% CI)
Relational	41	1	1	3	1	4	2	2	16	3	2	2	4	0.3 (0.1–0.7)	7.06	13.83 (NA)
Psychological	27	1	0	4	0	2	2	1	7	3	1	3	3	0.78 (NA)	1.38	1.98 (NA)
Physiological and health	15	0	1	1	0	1	2	0	6	0	1	0	3	2.68 (NA)		
Occupational	2	0	0	0	0	1	0	1	0	0	0	0	0	2.27 (NA)	29.03	55.8 (NA)
Sociodemographic	27	1	3	3	0	1	2	2	8	3	2	0	2	0.13 (NA)	1.25	2.37 (NA)
Predictors of response to trauma	16	0	2	0	0	1	2	0	7	0	2	0	2	-	-	-
Lifestyle	0	0	0	0	0	0	0	0	0	0	0	0	0	-	-	-
Negative environmental exposures	0	0	0	0	0	0	0	0	0	0	0	0	0	-	-	-
Genetic	0	0	0	0	0	0	0	0	0	0	0	0	0	-	-	-
Neuroanatomical	0	0	0	0	0	0	0	0	0	0	0	0	0	-	-	-
Total	128	3	7	11	1	10	10	6	44	9	8	5	14			

* Independence of these potential risk categories/factors cannot be ascertained with available data; ** odds ratio and relative risk of potential risk factors as indicated in the SRs/primary studies; NA, not available; CI-Confidence interval.

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
