# Peer review of "Perinatal Mental Illness in the Middle East and North Africa Region—A Systematic Overview"

_ijerph, 2020, doi:10.3390/ijerph17155487_

Round 1

Reviewer 1 Report

I applaud the very structured, systematic approach taken and described to generate the data. It is hereby nice to read that there is a significant amount of primary studies (n=134), but I agree that the ‘pooled’ meta-analysis is of add on value to further develop the topic and to provide guidance to this research line.

Disease labelling is a very relevant issue in perinatal mental illness, so additional information on the diagnostic tools (and its validity if based on questionnaires) as applied is very important. Although it is not part of the usual quality assessment, it can be a relevant factor for the prevalence results, so that I suggest the authors at least further reflect on this.

Table 4 summarized the various risk factors, but the reviewer assumes that these risk factors are likely not independent factors. I suggest to further reflect on this ?  

Specific comments and editing

Abstract:

SR is not explained as abbreviation.

Suggest to add the morbidity to the abstract, if there is sufficient word count to do so.  

Introduction,

line 44: what about subsequent mental illnesses in the offspring ? or epigenetic changes as mechanism (eg Hompes et al, J Psychiatr Res 2013).

Population of interest: any idea on the annual birth rate or another indicator for the number of pregnant women or deliveries in this population ?

Author Response

Response to reviewer comments:

Reviewer 1:

Point 1: I applaud the very structured, systematic approach taken and described to generate the data. It is hereby nice to read that there is a significant amount of primary studies (n=134).

Response 1: All authors thank the reviewer for this kind and encouraging remark.

Point 1.1: I agree that the ‘pooled’ meta-analysis is of add on value to further develop the topic and to provide guidance to this research line

Response 1.1: We agree with the reviewer that pooled meta-analysis would be highly valuable to further develop the topic and provide guidance for further research. Due to the extreme heterogeneity of the measurement tools and cut-offs used, we were unable to conduct a meta-analysis in our overview as they could have been misleading. We have elucidated this in the limitation section of the manuscript (lines 90-92 in page 17 of the manuscript).

Point 2: Disease labelling is a very relevant issue in perinatal mental illness, so additional information on the diagnostic tools (and its validity if based on questionnaires) as applied is very important. Although it is not part of the usual quality assessment, it can be a relevant factor for the prevalence results, so that I suggest the authors at least further reflect on this.

Response 2: We agree with this important observation of the reviewer. As per the reviewer’s suggestion, we now further reflect on this issue. Additional explanation has been added under ‘Quality assessment of the primary studies’ (lines 49-59 in page 12 of the manuscript).

Point 3: Table 4 summarized the various risk factors, but the reviewer assumes that these risk factors are likely not independent factors. I suggest to further reflect on this?  

Response 3: We thank the reviewer for this insightful comment. We have now clarified this further under ‘Overview of primary studies with data on risk factors’ (lines 142-146 in page 14 of the manuscript) and also in the footnote of Table 4 (Title: Number of studies exploring various potential risk factors for perinatal mental illness) that these factors are not necessarily independently associated with the studied outcomes.

Point 4: Specific comments and editing

Abstract:

Point 4.1: Point SR is not explained as abbreviation.

Response 4.1: Many thanks for this kind observation. We have now expanded SR (systematic review) in the abstract (line 12 of page 1 of the manuscript).

Point 4.2: Suggest to add the morbidity to the abstract, if there is sufficient word count to do so.  

Response 4.2: As per the reviewer’s valuable suggestion, we have now added ‘morbidity’ to line 10 of page 1 of the manuscript.

Introduction:

Point 4.3: line 44: what about subsequent mental illnesses in the offspring? or epigenetic changes as mechanism (eg Hompes et al, J Psychiatr Res 2013).

Response 4.3: We thank the reviewer for this kind suggestion. We have now reflected this in lines 43 and 44 of page 1 of the manuscript. The suggested reference has been also been duly added.

Point 4.4: Population of interest: any idea on the annual birth rate or another indicator for the number of pregnant women or deliveries in this population?

Response 4.4: We have now included the annual (crude) birth rate for the MENA region, estimated at 23/1000 persons as per the World Bank data of 2018 (lines 97-98 in page 3 of the manuscript).

Reviewer 2 Report

Abstract.

Please specify if those paper were the ones you found at the initial search or the ones you examined after the application of inclusion/exclusion criteria of the review?

Please avoid some abbreviations for the first time they are used (reader may not be completely familiar with them)

You should use Vancouver citations and reference system, so you have to change it through the manuscript. It´s so hard to read the manuscript because of the greater number of citations after each sentence. Once you make this change, please revise if the extension of the manuscript is ok.

Discussion

Please provide a specific paragraph regarding the implications of your manuscript

Author Response

Response to reviewer comments:

Reviewer 2:

Point 1: Abstract

Point 1.1: Please specify if those papers were the ones you found at the initial search or the ones you examined after the application of inclusion/exclusion criteria of the review?

Response 1.1: All authors thank the reviewer for this observation. We have now clarified in lines 21 and 22 of page 1 of the manuscript that the number of studies mentioned are the ones included after the application of inclusion/exclusion criteria of the review.

Point 1.2 Please avoid some abbreviations for the first time they are used (reader may not be completely familiar with them)

Response 1.2: This comment by the reviewer is duly noted.  We identified that we had missed out to expand SRs as systematic reviews in (line 12 of page 1) of the manuscript. We have now addressed it. We have also reviewed the manuscript fully to ensure that abbreviations are not used when they feature for the first time in the manuscript.

Point 2: You should use Vancouver citations and reference system, so you have to change it through the manuscript. It´s so hard to read the manuscript because of the greater number of citations after each sentence. Once you make this change, please revise if the extension of the manuscript is ok.

Response 2: We thank the reviewer for this observation and suggestion. We have now used the MDPI journals recommended reference format (very similar to the Vancouver style) in the manuscript.

Point 3: Discussion

Point 3.1: Please provide a specific paragraph regarding the implications of your manuscript

Response 3.1: We thank the reviewer for this suggestion. We have now included an exclusive paragraph under the discussion section (lines 74-87 in page 17 of the manuscript) regarding the implications of our manuscript.